# Hepatitis C Virus as a Possible Helper Virus in Human Hepatitis Delta Virus Infection

**DOI:** 10.3390/v16060992

**Published:** 2024-06-20

**Authors:** Maria Grazia Crobu, Paolo Ravanini, Clotilde Impaloni, Claudia Martello, Olivia Bargiacchi, Christian Di Domenico, Giulia Faolotto, Paola Macaluso, Alessio Mercandino, Miriam Riggi, Vittorio Quaglia, Stefano Andreoni, Mario Pirisi, Carlo Smirne

**Affiliations:** 1Laboratory of Molecular Virology, Maggiore della Carità Hospital, 28100 Novara, Italy; mcrobu@cittadellasalute.to.it (M.G.C.); paolo.ravanini@gmail.com (P.R.); clotildeimpaloni@gmail.com (C.I.); claudiamartello14@gmail.com (C.M.); christian.didomenico@maggioreosp.novara.it (C.D.D.); giulia.faolotto@maggioreosp.novara.it (G.F.); paola.macaluso@maggioreosp.novara.it (P.M.); alessio.mercandino@maggioreosp.novara.it (A.M.); miriam.riggi@maggioreosp.novara.it (M.R.); vittorio.quaglia@maggioreosp.novara.it (V.Q.); stefano.andreoni@maggioreosp.novara.it (S.A.); 2Clinical Biochemistry Laboratory, Department of Laboratory Medicine, AOU Città della Salute e della Scienza di Torino, 10126 Turin, Italy; 3Unit of Infectious Diseases, Maggiore della Carità Hospital, 28100 Novara, Italy; olivia.bargiacchi@maggioreosp.novara.it; 4Internal Medicine Unit, Department of Translational Medicine, University of Piemonte Orientale, 28100 Novara, Italy; mario.pirisi@med.uniupo.it

**Keywords:** hepatitis C virus, hepatitis D virus, hepatitis B virus, human immunodeficiency virus, propagation, anti-HBc, HDV-RNA, helper virus, co-infection, superinfection

## Abstract

Previous studies reported that the hepatitis C virus (HCV) could help disseminate the hepatitis D virus (HDV) in vivo through hepatitis B virus (HBV)-unrelated ways, but with essentially inconclusive results. To try to shed light on this still-debated topic, 146 anti-HCV-positive subjects (of whom 91 HCV/HIV co-infected, and 43 with prior HCV eradication) were screened for anti-HDV antibodies (anti-HD), after careful selection for negativity to any serologic or virologic marker of current or past HBV infection. One single HCV/HIV co-infected patient (0.7%) tested highly positive for anti-HD, but with no positive HDV-RNA. Her husband, in turn, was a HCV/HIV co-infected subject with a previous contact with HBV. While conducting a thorough review of the relevant literature, the authors attempted to exhaustively describe the medical history of both the anti-HD-positive patient and her partner, believing it to be the key to dissecting the possible complex mechanisms of HDV transmission from one subject to another, and speculating that in the present case, it may have been HCV itself that behaved as an HDV helper virus. In conclusion, this preliminary research, while needing further validation in large prospective studies, provided some further evidence of a role of HCV in HDV dissemination in humans.

## 1. Introduction

Hepatitis delta virus (HDV) is a defective human virus that lacks the ability to produce its own envelope proteins and, thus, depends on the presence of a helper virus, which induces its surface proteins to produce infectious particles. Until now, hepatitis B virus (HBV) was thought to be the only helper virus associated with HDV [1]. However, recent evidence has shown that divergent HDV-like viruses can be detected in fishes, birds, amphibians and invertebrates without evidence of any HBV-like agent to support co-infection [2]. Another recent study showed that HDV can be transmitted and disseminated in experimental ex vivo and in vivo infections by several enveloped viruses unrelated to HBV, which possess surface glycoproteins (GPs) able to package HDV ribonucleoproteins (RNPs). This allows the efficient exit of HDV particles into the extracellular milieu of co-infected cells and subsequent entry into the cells expressing the respective receptors. Mechanisms of this sort have, so far, been demonstrated for flaviviruses (such as dengue and West Nile virus), vesiculovirus and hepatitis C virus (HCV) [3]. Taken together, these results suggest that hepatitis D infection could, in theory, occur in patients who carry either virus. Further confirmation in that sense comes from research demonstrating that snake HDV can efficiently utilize co-infecting reptarena- and hartmaniviruses to form infectious particles; even the expression alone of the envelope proteins of each virus was sufficient to induce the production of infectious HDV particles [4].

As for in vivo studies, there is very little evidence in the literature so far. The first survey available in the literature specifically analyzing HDV co-infection in human subjects with no apparent concomitant HBV infection was conducted in South America (Venezuela), where a series of HCV-RNA-positive patients was analyzed for HDV infection prior to initiation of any antiviral treatment for HCV. Sera from 2 of 160 subjects (1.25%) were positive for HDV antibodies (anti-HD), tested by enzyme-linked immunosorbent assay and further confirmed by LIAISON-XL immunoassay, but were negative for HBV serologic or molecular markers [e.g., hepatitis B surface antigen (HBsAg), anti-hepatitis B core (HBc) antibodies and HBV-DNA assessed by quantitative real-time PCR (qPCR) and digital droplet PCR (ddPCR)]. One of these two samples was also HDV-RNA detectable at nested PCR; the PCR-purified fragments were then sequenced and revealed HDV-RNA of genotype 1, which is frequently found in that country. These observations provided preliminary evidence that HDV infection can occur in the absence of HBV infection and that HDV spread might happen in humans without HBV acting as a helper virus. Instead, HCV might have been responsible for the diffusion of HDV, although the precise temporal correlation between HDV and HCV co-infections cannot be deduced from the clinical data provided by the authors. Another possible explanation for this unusual pattern of HDV infection could have been co-infection with an occult HBV; however, in the cited study, a confirmatory liver biopsy was not performed in these two positive subjects [5].

No similar study has ever been conducted in Italy. Yet, this country has been central to the monitoring of the epidemiology of HDV infection throughout the last four decades. This is because it was the first among industrialized countries to introduce universal HBV vaccination for neonates and 12-year-old adolescents in 1991, so that by 2020, almost all 40-year-old Italians were protected from HBV and, by default, from hepatitis D. However, no epidemiological data on the presence of possible HDV co-infection (inferred through anti-HD detection) are available outside the setting of hepatitis B (in which case rates are reported decreasing from 24.6% in 1983 to 8.9% in 2022 for HBV-positive subjects, and from 28.1% in 1997 to 8.4% in 2022 for HBV/human immunodeficiency virus (HIV) positive patients) [6,7,8].

The aforementioned preliminary results and the lack of reliable epidemiological data prompted us to search for HDV co-infection in a series of Italian HCV-infected subjects.

## 2. Materials and Methods

In this observational multicenter retrospective case series, all available consecutive sera of anti-HCV-positive subjects were considered that had been collected in the Hepatology or Infectious Diseases outpatient departments of four hospitals located in an Italian region (Piedmont) and then stocked at −80 °C at a centralized virology laboratory (Novara University Hospital) for molecular HCV viral load testing. The recruitment period for this study was from January 2022 to July 2023. In order to intercept subjects at higher risk for parenterally transmitted viral infections also HCV patients co-infected with HIV were considered. Specifically, inclusion criteria were: (a) positivity for markers of HCV infection (anti-HCV antibodies ± HCV-RNA) in the presence or absence of HIV infection (i.e., anti-HIV antibodies ± HIV-RNA); (b) patient age ≥ 18 years and written informed consent to participation in the study, (c) available medical records containing thorough clinical and laboratory information. Exclusion criteria were: (a) positivity of any markers of current or past HBV infection [i.e., HBsAg, hepatitis B e antigen (HBeAg), anti-HBc antibodies, anti-hepatitis e antigen (HBe) antibodies, HBV-DNA]; (b) any current anti-HCV treatment; (c) lack of a sufficient amount of sample; and (d) first diagnosis of known HCV positivity from less than a year before the sample collection used for this study. In HCV-positive subjects in whom HBV serology was not known, the latter one was performed on the sample selected for this study, prior to any patient enrollment. HCV-RNA-positive patients who did not have a liver elastography (Fibroscan^®^) performed within six months of blood collection were recalled for testing. All subjects gave written informed consent to their participation in the protocol, which was conducted in strict adherence to the principles of the Declaration of Helsinki of 1975, as revised in 2000. Researchers decided to investigate also a further patient, originally not included in the protocol (being HBV-positive) as better detailed in the Results section, deeming it crucial to draw further conclusions on the present investigation; since this subject was already deceased when the study was conducted, written consent for publication of his anonymized clinical details was collected by next of kin (wife). The study protocol was approved by the local institutional Ethics Committee (Comitato Etico Interaziendale Novara, https://comitatoetico.maggioreosp.novara.it/, (accessed on 1 June 2024), IRB code CE033/2023). Concerning the aforementioned additional patient, ethical approval was granted exemption both because it was considered as a case report and all the procedures being performed were part of the routine care and not a formal research protocol.

The test used for anti-HD detection was HDV Ab—ELISA (Dia.pro Diagnostic Bioprobes, Sesto San Giovanni, Italy), a competitive enzyme immunoassay with the semiquantitative concentration of anti-HD specific IgG antibodies determined by means of the ratio between a cut-off value (Co)—obtained with the following formula: [optical density (OD) of the negative control/OD of the positive control]/5—and the sample OD at 450 nm (S). This Co/S ratio was interpreted as follows: <0.9 (negative: patient not considered infected from HDV); 0.9–1.1 (equivocal: patient needing to be retested on a second independent sample 1–2 weeks after first blood test); and >1.1 (positive: patient considered to have previous or present HDV infection). The other main serological and molecular assays employed in this study were: for HDV-RNA detection, HDV Real-TM Quant (Sacace Biotechnologies, Como, Italy) with a lower limit of quantitation of 30 copies/mL; for HCV-RNA detection, Alinity m HCV (Abbott Laboratories, Abbott Park, IL, USA) with a lower limit of quantitation of 12 IU/mL; for anti-HCV antibody detection, LIAISON^®^ XL MUREX HCV Ab (DiaSorin, Saluggia, Italy); for HBV-DNA detection, Alinity m HBV (Abbott Laboratories) with a lower limit of quantitation of 10 IU/mL; for HBsAg detection, LIAISON^®^ XL MUREX HBsAg Quant (DiaSorin); for anti-HBsAg antibody (anti-HBs) detection, LIAISON^®^ XL MUREX Anti-HBs & Plus (DiaSorin); for HBc antibody detection, LIAISON^®^ Anti-HBc (DiaSorin); for HBeAg detection, LIAISON^®^ HBeAg (DiaSorin); for HBe antibody detection, LIAISON^®^ Anti-HBe (DiaSorin); for HIV-RNA detection, Alinity m HIV-1 (Abbott) with a lower limit of quantitation of 20 copies/mL; and for anti-HIV antibody detection, LIAISON^®^ XL MUREX HIV Ab/Ag (DiaSorin).

For what concerns statistical analysis, continuous variables were expressed as means, medians and ranges, and categorical variables as percentages. The Mann–Whitney, Wilcoxon, and Kruskal–Wallis tests were used to compare continuous non-parametric variables, as appropriate. Pearson’s chi-squared test was used to determine whether there was a significant difference between the expected and the observed frequencies in one or more categories. A *p* value of <0.05 was considered to be significant. All analyses were performed using Statistica 10.0 statistical software (Statsoft, Palo Alto, CA, USA).

## 3. Results

### 3.1. Whole Study Population

For this study, 705 anti-HCV subjects were screened; of these, 251 were tested also for HBV serology because it was lacking or incomplete. The following patients were excluded from the study after applying the aforementioned criteria: known positivity for any markers of current or past HBV infection (n = 272); positivity for one or more HBV markers found after screening test for subjects with not known or incomplete HBV serology (n = 105); current anti-HCV treatment with direct-acting antiviral agents (DAA) (n = 18); first HCV diagnosis from less than one year before study entry (n = 9); lack of a sufficient amount of plasma sample (n = 112); and lack of available complete medical records (n = 43). Ultimately, 146 anti-HCV patients (73 M, 73 F) with a median age of 59 years [interquartile range (IQR) 54–71 years] were selected for this study: 55 were HCV mono-infected, while the remaining 91 were HCV/HIV co-infected. No significant demographic differences were found between the two subgroups, except for age and risk factors for HIV co-infection [the latter ones being more frequently younger and intravenous drug users (IVDUs)]. The two subpopulations were also comparable for what concerns the main clinical parameters, even though HCV mono-infected persons showed a trend towards some more liver damage, possibly due to the fact that there was a trend for a lower proportion of subjects in this subpopulation who had already achieved a sustained virological response (SVR) at the study entry. Specifically, HCV mono-infected patients generally showed higher levels of aminotransferases and bilirubin, coupled with a somewhat worse liver biosynthetic capacity as assessed by serum albumin and international normalized ratio for prothrombin time (INR). However, a median model for end-stage liver disease (MELD) score—the most validated chronic liver disease severity scoring system that uses patient laboratory values for serum bilirubin, creatinine and INR—was perfectly comparable between the two subpopulations. Concerning SVR, in the whole studied population of 43 subjects (out of 45 treated, a success rate of 96%) had eradicated HCV during previous antiviral treatments [i.e., pegylated (PEG) interferon ± ribavirin and/or DAA], consequently resulting HCV-RNA negative when tested for this study. Of the anti-HIV-positive subjects, all were under antiretroviral treatment (HAART) when recruited in this study. The main characteristics of the studied population are presented in Table 1. All patients were tested for IgG anti-HD, and resulted negative except one.

### 3.2. Subanalysis of the Subject Found Positive for Anti-HDV Antibodies

The only anti-HD positive subject in our population was a HCV/HIV co-infected Caucasian 56-year-old female; HDV strong positivity (S: 0.062, Co/S ratio: 9.484) was then confirmed in a second independent sample obtained two weeks after the first blood test (S: 0.075, Co/S ratio: 8.120). She was also tested for HBV-DNA, which resulted negative both at qualitative and quantitative detection. In addition, the sample was also checked for the presence and quantification of HDV-RNA, which turned out to be negative.

The subject was a previous IVDU, specifically heroin from 1982 to 1999, with also a persistent moderate alcohol consumption (3 drinks/day) and a history of smoking (30 packs/year). Her HIV diagnosis dated back to 1993, following the discovery of the infection in her husband. In the following year, during her first pregnancy, she was also found to be HCV-positive (with high viral load, HCV genotype 1a and interleukin (IL)-28b genotype C/T) [9]. Her other relevant comorbidities were obesity (BMI 30 kg/m^2^) and hypertension medically treated. For what concerns chronic hepatitis C, she was always demonstrated to have mild liver fibrosis (METAVIR stage 1 to repeated liver elastography examinations), and she eventually reached SVR with a DAA antiviral treatment (i.e., sofosbuvir/velpatasvir) after a previous failure with a three-month regimen with PEG α2a + ribavirin (stopped because of a virological null response). As expected, for her HIV disease, she was given a variety of HAARTs. Specifically, she decided to start antiviral treatment at the end of 1999, during her second pregnancy; at that time, her HIV-RNA setpoint was 60,000 copies/mL. Since then, she was almost always on therapy, but, especially in the first years, with problems of poor compliance, partially motivated by drug side effects (e.g., allergy to abacavir and intolerance to ritonavir) and number of pills. This resulted in the development of resistance to some compounds and stably detectable HIV viremias. After that, in the last 14 years (since the second half of 2009), she finally achieved excellent compliance and stable virological suppression. Her current antiviral therapy (which was already present when she was recruited for this study) consists of dolutegravir plus darunavir/cobicistat (started three years ago), but in the past, some regimens included drugs, alone or in combination with known anti-HBV activity, such as lamivudine, tenofovir or emtricitabine. In any case, her HBV markers (both complete serology and viremia) have always been completely negative from her first diagnosis of HIV infection more than 30 years ago. A deeper analysis of the demographic and clinical features of this patient is presented in Table 2 panel a, while a timetable reporting the whole scheme of her anti-HIV and anti-HCV treatments, as well as HBV screening tests, is reported in Figure 1.

### 3.3. Subanalysis of the Anti-HDV Positive Patient’s Partner

In order to better understand the possible source of her HDV transmission, we decided to study also her husband, who was initially excluded from this study protocol as he had known previous contact with HBV (in other words, a main exclusion criterion). He, too, was a former IVDU (heroin from 1983 to 1999), with a known HIV infection from 1986 and persistent moderate alcohol consumption (4 drinks/day) and a history of heavy smoking (40 packs/year). Since 1996, he had been treated with several antiviral treatments (pre-therapy HIV-RNA setpoint: 60,000 copies/mL), the last of which was analogous to his partner (i.e., dolutegravir plus darunavir/cobicistat). During the first years of treatment, he experienced severe problems of compliance, partially due to social problems (he was a prisoner for a number of years), leading to the emergence of multiple resistances and stably positive HIV-RNA. However, in the last eight years, he achieved excellent immunovirological compensation (with CD4+ T-cell count greater than 0.60 × 109/L and persistently undetectable viremias). As a major comorbidity, the patient in 1985 had a non-A non-B acute hepatitis, followed by a diagnosis of a genotype 1a chronic hepatis C (likewise his partner, with high viral load and IL-28b genotype C/T), which evolved to well-compensated cirrhosis [on average, Child–Pugh score: A5, model for end-stage liver disease-sodium (MELD Na) score: 7]. He finally achieved an SVR after a DAA treatment with sofosbuvir/ledipasvir plus ribavirin, following a failure with a six-month regimen with PEG α2a + ribavirin. Different from his partner, as previously reported, he also had a history of previous contact with HBV. Since we were able to retrieve blood test results from the electronic archive of our hospital laboratory (mid-2006) he was always anti-HBc positive, with both HBsAg and circulating HBV-DNA negative [thus to be considered, until proven otherwise, as a subject with a seropositive occult HBV infection (OBI)] [10,11]. In this respect, it must be said that this subject, like his wife, experienced multiple past antiretroviral therapies with concomitant anti-HBV activity, such as lamivudine, tenofovir and emtricitabine. Interestingly, seroconversion to both anti-HBs and anti-HBe was detected at the last available HBV test (late 2020), without any concomitant liver enzyme elevation, as already reported by others during prolonged HAARTs [12,13]. He eventually died one year before the beginning of the present study (mid-2021) as a result of a locally advanced lung adenocarcinoma treated with various lines of chemotherapy and immunotherapy. Regarding hepatitis D, he was never tested for the presence of delta antigen, anti-HD, or HDV-RNA. Table 2 panel b reports a thorough analysis of the last available demographic and clinical features of this subject before the cancer diagnosis (late 2020, when he was 57 years old), while a scheme of his anti-HIV and anti-HCV treatments, as well as HBV tests, is reported in Figure 2.

## 4. Discussion

Hepatitis delta is the most aggressive form of viral hepatitis and is caused by HDV, the smallest human RNA virus. It is well known that HDV usually requires the simultaneous presence of HBV to complete its life cycle. As a consequence, infection by HDV occurs through two main modes, namely as simultaneous infection (co-infection) with both HBV and HDV or as a superinfection by HDV of a chronic HBV carrier [14]. A third possible mode of HDV infection refers to HDV mono-infection of hepatocytes, with host RNA polymerases replicating HDV-RNA in the absence of HBV infection, which is not productive but can be maintained by HBV helper virus at a later stage [14,15,16,17,18,19,20]. In humans, HBV-independent HDV replication has so far exclusively been demonstrated in the salivary glands of patients affected by Sjögren’s syndrome (suggesting, incidentally, a possible non-hepatic reservoir for chronic HDV persistence) [21,22] and, for a short period, in the context of liver transplantation [23,24].

HDV causes more severe liver disease than HBV alone and is associated with accelerated liver fibrosis, liver cancer and liver failure. It is a condition with a significant impact on global health, potentially affecting up to about 15–20 million people worldwide. The prevalence of HDV varies in different parts of the world, with the highest reports in the Mediterranean basin, the Middle East, Central and North Asia, West and Central Africa, the Amazon basin, the Pacific islands, and Vietnam [25,26]. Regardless of geographic areas, amongst HBsAg-positive persons, higher anti-HD rates are found in IVDUs, hemodialysis recipients, men who have sex with men, commercial sex workers, and those with HCV or HIV co-infections [27,28,29,30]. Despite its severity, HDV infection screening remains inadequate. Detection of anti-HD antibodies should be the first step in the diagnosis, as they can be detected at high titers in superinfected patients and at lower levels in co-infections. However, two main limitations must be kept in mind, which justify the need for complementary approaches to confirm the diagnosis. First, total anti-HDV antibodies may be undetectable in the first weeks of acute infection. Second, anti-HDV IgG may persist after HDV infection, not allowing to distinguish between active and resolved infection. Confirmation of infection is thus based on the detection of HDV-RNA by quantitative reverse transcriptase (RT)-PCR, which—together with the positivity of anti-HDV antibodies—makes it possible to distinguish between chronic and previous infections and to follow the response to treatments [31].

In this study we investigated if an enveloped virus distinct from HBV, namely HCV, may induce dissemination of HDV in vivo. Our research, which has the merit of being conducted in a real-world setting, enabled us to confirm previous preliminary evidence for HDV exposure in HCV patients who were apparently not infected with HBV. In our population the prevalence of anti-HD was 0.7%, comparable to the very few in vivo experiences so far reported in the Literature, both dating to 2021.

The first one is the South American study described above (anti-HD prevalence around 1%) [5]. Nevertheless, this investigation also includes many potential biases, aside from the lack of histological confirmation already reported. First of all, it did not test for all HBV serological markers; furthermore, the precise methodology of the study and why patients were collected in two non-consecutive periods (2004–2005 and 2014–2015) are unclear; finally, the study is very poor in terms of clinical and demographic characterization of the tested subjects (e.g., key variables such as gender, age, comorbidities are not provided; HIV co-infection status is not clearly stated, even though it was presumably negative). However, the research has undoubted merits, such as that of having been conducted on subjects all with active HCV infection (i.e., HCV-RNA positive) and having found at least one individual with positive HDV-RNA.

To the best of our knowledge, there is currently only one other comparable study, which was published a few months later, but in another area of the world (France). This is a research study on 2123 anti-HCV antibody-positive (and HIV-negative) plasma samples, reporting a higher prevalence of anti-HD positivity (1.9%). However, of the 41 anti-HD-positive subjects, 27 (65.9%) were anti-HBc positive. As a consequence, anti-HD was significantly more present in samples positive for anti-HBc (6.21% vs. 0.8%; *p* < 0.001). Focusing on the 14 anti-HD positive but anti-HBc negative subjects, they were mainly HCV-RNA negative (9 vs. 5). Interestingly, the anti-HD Co/S ratios observed were much higher in anti-HBc positive (median: 4.40) than in anti-HBc negative samples (median: 1.48, *p* < 0.01). In comparison, in our research, the only anti-HD positive patient had an even greater C0/S ratio (though obtained with a different laboratory method), even if she was anti-HBc negative. Similar to our study, in the French cohort, no anti-HD positive samples were also found with detectable HDV-RNA (but this investigation could be conducted on only 28/41 anti-HD positive subjects). Although this study is intriguing and has a good sample size, again, it incorporates some big methodological problems, the main being that is was centered on the anti-HBc status. In this regard, the Methods section reports that HBV active co-infections (i.e., HBV-DNA or HBsAg positive subjects) were excluded, but no clear mention is made of the other HBV serological markers (such as anti-HBs or anti-HBe). Moreover, being research conducted on blood donors, there was obviously no possibility of histological liver investigations that could have ruled out an OBI [32].

In any case, all these findings, together with our results, confirm the possibility of HCV as a new helper virus for HDV in vivo. It must be said that in the literature, there is also a third study on HCV-infected subjects negative for HBsAg, which found an even higher prevalence (8/323, 2.5%) of anti-HD positive subjects (again, none with positive HDV-RNA). The problem is that all the anti-HD positive subjects were also found positive for anti-HBc. So, as this is an obvious sign of previous HBV infection, this study, unlike the previous ones, cannot give any additional information on a hypothetical role of HCV as a helper virus independent of HBV [33].

Focusing on our analysis, some additional considerations deserve to be made, the most important being that, as reported in the Methods section, we looked for possible HDV infection not only in mono-infected HCV-positive patients but also in subjects co-infected with HIV, the latter ones in order to investigate persons presumably at higher risk for parenterally transmitted infections. Moreover, we aimed to investigate a model (i.e., HIV/HDV co-infected patients) in which, notoriously, liver disease can progress more rapidly than in HIV mono-infected individuals, regardless of the use of successful HAART [34,35]. Now—as far as we know—there is no study that formally investigated the real burden of HDV superinfection or co-infection in the subset of HCV/HIV-positive patients, so our research is unprecedented in literature. As a matter of fact, the few available data concerning the prevalence and epidemiology of hepatitis D amongst HIV-positive subjects refer to subjects with known HBV co-infection. Specifically, in the large EuroSIDA cohort of HIV-infected individuals, the overall prevalence of anti-HD antibodies was 14.5% [28]. Reported new rates in Europe were higher in Southern and Eastern countries than in Central and Northern ones, with most new diagnoses among immigrants from HDV-endemic regions, while in the 1980s IVDUs were the most prevalent category anywhere [36,37,38]. In other areas, such as Taiwan, the rate of delta hepatitis was also higher in the HIV/HBV co-infected population than among individual HBsAg carriers [30].

Coming back to HCV/HIV co-infections, it is noteworthy that our only HDV-positive case belonged to this specific category. From the elements we have available, it can be assumed that the patient contracted HDV without concurrent HBV infection. Clinical data show instead that her partner was positive for both HBV and HCV. We can thus speculate that the anti-HBV therapy given in this latter patient may have reduced HBV viral replication for a time while maintaining the possible replication of any associated HDV; he would then have had the possibility of transmitting HDV independently of HBV. The recipient of the infection (in this case, our patient who resulted HDV-positive) may have supported HDV replication precisely because of her HCV-positive status (i.e., with circulating HCV-RNA). The rationale for that comes from an elegant in vitro model by Perez-Vargas, where the co-transfection of the Huh-7 cell line with two plasmids (one coding for HDV RNPs—which in turn are in close association with HDV-RNA— and the other for HCV GPs) allowed the packaging of HDV RNPs in the HCV-envelope GPs (i.e., E1E2). This, in turn, prompted the release of HDV particles into the extracellular milieu by the infected cells. In addition, the same group provided in vivo evidence that HCV could continue to spread HDV infection in the liver of co-infected humanized mice for several months. The authors demonstrated that in these animals -as in the case of enveloping with HBsAg-it was the farnesylation of HDV RNPs that was necessary for assembling HDV particles with the HCV coat [3]. These results indicate that farnesyl-mediated targeting to endoplasmic reticulum or other cell membranes is required for assembly of the HCV-induced HDV particles (HCV-Δp), suggesting that they share with HDV the same early steps and lead to the production of infectious particles [3,39,40]. As a further confirmation, when inhibiting this pathway by lonafarnib, a farnesyltransferase inhibitor, the production and transmission of both HDV and HCV-Δp were promptly reduced [41]. In other words, there is some robust experimental evidence that HCV may be responsible for the spread of HDV, although it was not possible to infer from our available clinical data whether HDV infection preceded, was simultaneous with, or occurred after HCV co-infection.

It must be said that in our sample, despite the high positive anti-HD titer, HDV-RNA could not be detected, which means that HDV, at the time of collection, was not in an active replicative phase [42]. This situation may be explained by the previous eradication of HCV as a result of the antiviral therapy with DAA; in other words, SVR in this situation would imply the simultaneous total suppression of HDV replication as well. All that would remain of this infection, therefore, would be IgG anti-HD. From a speculative point of view, there would also be another possible explanation for this unusual pattern of HDV infection, that is, a co-infection with a seronegative OBI, with both serological markers and circulating HBV-DNA being negative [11,43,44,45]. However, this event is extremely rare: to rule out this chance, the only known way would be to perform an HBV-DNA test on a liver biopsy. As it was not possible to perform this investigation, this represents the obvious most important limitation of our research. However, it must be said that the few comparable studies with ours share the same defect [5,32]. Another possible drawback may be that the patient’s partner was not tested for HDV markers, which makes the aforementioned hypothesis of infectious transmission plausible, but not definitely confirmed.

A theoretical alternative explanation could also be HIV-HDV co-infection. In this regard it must be said that the patient had detectable HIV viremia for many years, both due to her delay in starting HAART and subsequent problems of poor compliance, as described above. However, it is the authors’ opinion that this is a speculation, which at the moment seems very unlikely. First of all, to the best of our knowledge, there is currently no experimental evidence that HIV could favor the replication of HDV, as demonstrated for other viruses. Although there is some recent preliminary evidence that the two viruses, though very different, may share some common molecular pathways. As a matter of fact, an unexpected proviral activity of JAK1 was reported both in the context of infections from HIV (using a JAK1-specific small interfering RNA (siRNA) in HeLa-derived cell lines) and HDV (using various in vitro models, including primary human hepatocytes). Yet, while during HIV infection the proviral role of JAK1 was associated with the activation of signal transducer and activator of transcription (STAT) 1 and STAT3 proteins, promoting enhanced inflammation and viral replication, in HDV models JAK1 promoted replication independent of STAT3 activation [46,47,48]. In any case, it must be said that currently there is no evidence that an enveloped virus like HIV may induce any dissemination of HDV in vitro, operating as a helper virus. Specifically, when applying the same model previously cited, HDV-RNA replication was not prevented, but no secretion of HDV particles could be detected in the cell culture supernatants, which, consequently, did not show any infectivity on target cells (i.e., Huh-106 cells) [3]. So, unlike other enveloped viruses that were demonstrated to be able to induce an efficient release of HDV particles by forming nucleocapsid-free subviral particles (SVPs) coated with envelope GPs (e.g., as previously mentioned, vesiculovirus, arenavirus, metapneumovirus and flaviviridae such as dengue virus, West Nile virus and HCV), the GPs of retroviruses such as HIV are indeed released from the plasma membrane or late endosomes upon incorporation onto the surface of infectious virions, but have not been demonstrated to form SVPs, which may precisely explain their failure to assemble viable HDV particles [3,49].

## 5. Conclusions

Given the worldwide endemicity of HCV, the obvious question is whether HDV-RNA can be transmitted to humans via HCV envelope GPs. Moreover, HDV and HCV share similar routes of transmission, mainly through parenteral and sexual exposure, increasing the at least theoretical likelihood that the two viruses may infect—simultaneously or sequentially—a proportion of patients with complex viral replication patterns and immune responses.

Our observation provides preliminary proof of HDV exposure in patients who were chronically HCV-infected without evidence of ongoing or past HBV infection, suggesting that the transmission of HDV through HBV-unrelated viruses in humans may be possible, albeit rare. Belonging to groups known to be at increased risk of acquiring parenterally transmitted infections (such as HIV-positive individuals) could possibly increase that chance, although large prospective confirmatory studies are needed before systematic screening for HDV might be recommended, at least in this very specific subset of HCV-positive individuals.

## Figures and Tables

**Figure 1 viruses-16-00992-f001:**
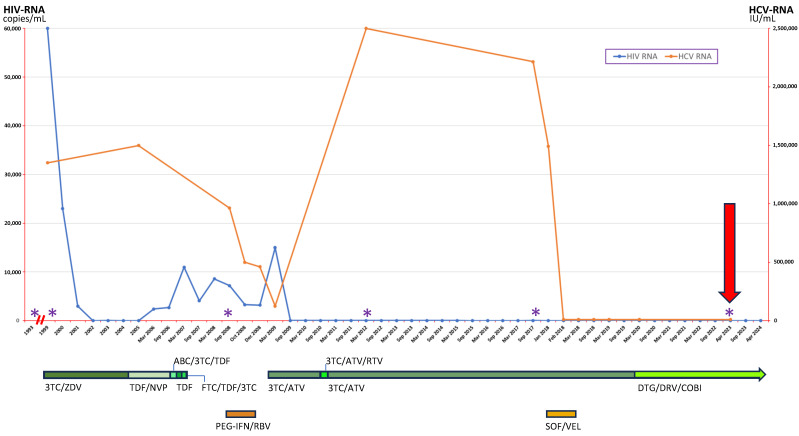
Timeline with relevant data from the clinical history of the patient, which resulted anti-HD positive. The upper part of the figure depicts data with respect to HIV-RNA (blue line) and HCV-RNA (orange line) trends; the purple asterisks represent the serialized controls of virological tests for HBV (HBV-DNA and complete serology, which were always completely negative). The lower part of the figure depicts the treatment regimens she underwent for both HIV (in green) and HCV (in orange) infections. The arrow refers to the time when the patient was sampled for the present study (April 2023), as reported in Table 1. Abbreviations: abacavir (ABC); atazanavir (ATV); cobicistat (COBI); darunavir (DRV); dolutegravir (DTG); emtricitabine (FTC); hepatitis B virus (HBV); hepatitis C virus (HCV); human immunodeficiency virus (HIV); lamivudine (3TC); nevirapine (NVP); pegylated interferon (PEG-IFN); ribavirin (RBV); ritonavir (RTV); sofosbuvir (SOF); tenofovir disoproxil fumarate (TDF); velpatasvir (VEL); zidovudine (ZDV).

**Figure 2 viruses-16-00992-f002:**
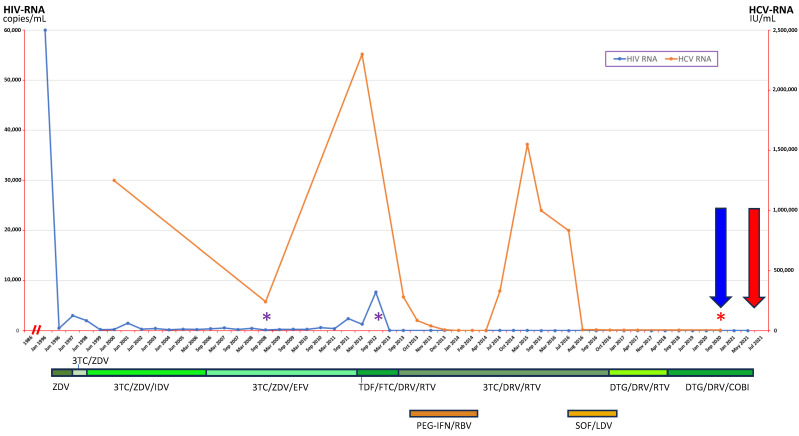
Timeline with relevant data from the clinical history of the anti-HD positive patient’s husband. The upper part of the figure depicts data with respect to HIV-RNA (blue line) and HCV-RNA (orange line) trends. Asterisks represent the serialized controls of virological tests for HBV (HBV-DNA and complete serology); color code: purple means completely negative serology, red means seroconversion to both anti-HBs (73 IU/mL) and anti-HBe (weak positivity). The lower part of the figure depicts the treatment regimens he underwent for both HIV (in green) and HCV (in orange) infections. The blue arrow refers to the last blood biochemical workup immediately prior to the diagnosis of lung cancer and that was considered for this research as reported in Table 1 (September 2020), while the red arrow refers to the patient’s date of death (July 2021). Abbreviations: anti-hepatitis e antigen antibodies (anti-HBe); anti-HBsAg antibodies (anti-HBs); cobicistat (COBI); darunavir (DRV); dolutegravir (DTG); efavirenz (EFV); emtricitabine (FTC); hepatitis B virus (HBV); hepatitis C virus (HCV); human immunodeficiency virus (HIV); indinavir (IDV); lamivudine (3TC); ledipasvir (LDV); pegylated interferon (PEG-IFN); ribavirin (RBV); ritonavir (RTV); tenofovir disoproxil fumarate (TDF); zidovudine (ZDV).

**Table 1 viruses-16-00992-t001:** Baseline characteristics of the studied population. Data are presented as median (range) for continuous variables and as frequency (%) for categorical variables. Bold values denote statistical significance at the *p* < 0.05 level.

	All Subjects (n = 146)	HCV Mono-Infected (n = 55)	HCV/HIV Co-Infected (n = 91)	HCV vs. HCV/HIV *P*	Local Laboratory NR
Sex, n (M, F)	73 (50), 73 (50)	22 (40), 33 (60)	51 (56), 40 (44)	0.060	-
Ethnicity, n (Caucasian, other)	143 (98), 3 (2) ^1^	55 (100), 0 (0)	88 (97), 3 (3) ^1^	0.317	-
Risk factor for HCV ± HIV acquisition, n (IVDU, sex, transfusion, other/not known)	73 (50), 4 (19), 3 (2), 33 (29)	15 (27), 4 (7), 3 (6), 33 (60)	58 (64), 24 (26), 0 (0), 9 (10)	**<0.001**	-
Age, years	59 (54–71)	71 (57–80)	57 (53–62)	**<0.001**	-
Liver elastography, KPa ^2^	7.6 (5.9–13.1)	8.5 (5.8–15.4)	7.3 (5.9–10.0)	0.522	<5.0
HCV RNA, ×10^3^ IU/mL ^2^	868 (320–2396)	782 (272–2912)	885 (385–2235)	0.841	negative
HCV genotype, n (1a, 1b, 2, 3, 4) ^3^	52 (36), 25 (17), 37 (25), 21 (14), 11 (8)	20 (36), 8 (15), 13 (24), 10 (18), 4 (7)	32 (35), 17 (19), 24 (26), 11 (12), 7 (8)	0.853	-
HIV RNA, copies/mL ^4^	-	-	0 (0–0)	N/A	negative
White blood cells, n × 10^9^/L	5.18 (4.23–6.28)	5.11 (4.08–6.05)	5.78 (4.69–7.55)	0.134	4.50–11.00
CD4+ lymphocytes, n × 10^9^/L ^4^	-	-	0.37 (0.27–0.86)	N/A	0.49–1.70
Platelets, n × 10^9^/L	199 (83–233)	182 (85–179)	202 (161–233)	**<0.001**	150–450
AST, IU/L	45 (26–98)	60 (37–140)	15 (13–18)	**<0.001**	0–40
ALT, IU/L	43 (22–103)	69 (33–119)	20 (16–23)	**<0.001**	0–40
Total bilirubin, mg/dL	0.8 (0.6–1.1)	0.8 (0.6–1.1)	0.5 (0.4–0.9)	**0.003**	0.30–1.20
Creatinine, mg/dL	0.77 (0.64–0.90)	0.80 (0.70–0.90)	0.77 (0.64–0.90)	0.992	0.60–1.10
INR, Units	0.9 (0.9–1.1)	1.1 (1.0–1.2)	1.0 (0.9–1.1)	**<0.001**	0.80–1.20
Albumin, g/L	39 (36–43)	38 (35–41)	45 (42–47)	**<0.001**	34–48
MELD, score	7 (6–8)	7 (6–8)	6 (6–7)	0.059	-
SVR after previous HCV treatments, n	43 (29)	13 (23)	30 (33)	0.231	-
Current HIV therapy, n ^4,5^	-	-	91 (100)	N/A	-
Current or past HIV therapies with concomitant anti-HBV effect, n ^4,6^	-	-	91 (100)	N/A	-

Abbreviations: alanine transaminase (ALT); aspartate transaminase (AST); bictegravir (BIC); cabotegravir (CAB); cobicistat booster (COBI); darunavir (DRV); dolutegravir (DTG); emtricitabine (FTC); hepatitis B virus (HBV); hepatitis C virus (HCV); human immunodeficiency virus (HIV); international normalized ratio (INR); intravenous drug user (IVDU); lamivudine (3TC); model for end-stage liver disease (MELD); normal range (NR); not applicable (N/A); raltegravir (RAL); rilpivirine (RPV); ritonavir booster (r); sustained virological response (SVR); tenofovir alafenamide (TAF); tenofovir disoproxil fumarate (TDF). ^1^ 1 Black, 1 Asian, 1 Hispanic subject. ^2^ For the 103 patients who were HCV-RNA positive when entering the study. ^3^ For all patients, including the 43 subjects who achieved SVR before entering the study. ^4^ For the 91 HCV/HIV positive patients. ^5^ BIC/FTC/TAF (n = 24), 3TC/DTG (n = 18), RAL/DRV/COBI (n = 13), DTG/DRVr (n = 8), TAF/FTC/DRV/COBI (n = 7), DTG/RPV (n = 6), DTG/DRV/COBI (n = 6), CAB/RPV (n = 5), TDF/FTC/RPV (n = 4). ^6^ Including one or more of the following ones: 3TC, TDF, TAF, FTC.

**Table 2 viruses-16-00992-t002:** Main laboratory findings of the anti-HDV positive subject (a) and of her husband (b). Local laboratory normal ranges are reported in (c): values outside those ranges are reported in bold.

	(a)	(b)	(c)
HCV RNA, IU/mL	Negative ^1^	Negative ^2^	negative
HIV RNA, copies/mL	**<20**	negative	negative
HBV DNA, IU/mL	negative	negative	negative
HDV RNA, copies/mL	negative	N/A	negative
HCV-Ab	**positive**	**positive**	negative
HIV-Ab	**positive**	**positive**	negative
HBsAg/HBsAb/HBcAb IgG/ HBcAb IgM/HBeAg/HBeAb	neg/neg/neg/neg/neg/neg	<0.0 IU/mL ^3^/**73 IU**/**mL**/**pos**/neg/neg/**weakly pos** ^4^	all negative
anti-HDV IgG	**positive**	N/A	negative
Liver stiffness at Fibroscan, Kpa	**6.5** ^5^	**15.6** ^5^	<5.0
Liver steatosis at Fibroscan (CAP), db/sq m	**277** ^5^	211 ^5^	<214
White blood cells, n × 10^9^/L	5.20	6.01	4.50–11.00
Neutrophil/lymphocyte count, n × 10^9^/L	2.82/1.90	3.73/1.56	1.80–7.70/1.00–4.50
CD4+ T lymphocytes, n × 10^9^/L	0.72	**0.44**	0.49–1.70
Hemoglobin, g/L	123	137	117–157
Platelets, n × 10^9^/L	215	264	150–450
AST, IU/L	19	19	0–40
ALT, IU/L	12	30	0–40
GGT, IU/L	13	164	0–50
ALP, IU/L	220	338	90–360
Total bilirubin, mg/dL	0.50	0.80	0.30–1.20
Creatinine, mg/dL	0.59	0.60	0.60–1.10
eGFR, mL/min	104	112	>90
Na^+^, mEq/L	138	141	134–146
Glucose, mg/dL	**106**	92	70–100
INR, Units	1.02	0.98	0.80–1.20
Albumin, g/L	47	42	34–48
Total cholesterol, mg/dL	**232**	**282**	0–200
Triglycerides, mg/dL	**241**	138	45–170

Abbreviations: alanine aminotransferase (ALT); alkaline phosphatase (ALP); anti-hepatitis B core antibodies (HBcAb); antibodies (Ab); anti-hepatitis e antigen antibodies (HBeAb); aspartate transaminase (AST); controlled attenuation parameter (CAP); direct-acting antiviral agents (DAA); estimated glomerular filtration rate (eGFR); gamma-glutamyltransferase (GGT); hepatitis B virus (HBV); hepatitis B e antigen (HBeAg); anti-hepatitis B surface antibodies (HBsAb); hepatitis B surface antigen (HBsAg); hepatitis C virus (HCV); hepatitis D virus (HDV); international normalized ratio (INR); liver stiffness (LS); not available (N/A); sodium (Na^+^). ^1^ data before DAA treatment: HCV RNA 2,216,000 IU/mL, HCV genotype 1a. ^2^ data before DAA treatment: HCV RNA 1,665,000 IU/mL, HCV genotype 1a. ^3^ quantitative detection. ^4^ available previous determinations: August 2008 HBV-DNA neg, HBsAg neg, **HBcAb IgG pos**, HBsAb 5 IU/mL, HBeAg neg, HBeAb neg; March 2012 HBV DNA neg, HBsAg neg, **HBcAb IgG pos**, HbsAb 0 IU/mL, HBeAg neg, HBeAb neg. ^5^ measured before DAA treatment.

## Data Availability

All data generated or analyzed during this study are included in this published article.

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
