# Peer review of "Hepatitis C Virus as a Possible Helper Virus in Human Hepatitis Delta Virus Infection"

_viruses, 2024, doi:10.3390/v16060992_

Round 1
Reviewer 1 Report
Comments and Suggestions for Authors
This is an interesting paper that presents a case of possible transmission of HDV and ongoing replication in humans using HCV as a helper virus instead of HBV.
Unfortunately proof is lacking (as authors acknowledge) as HDV RNA was not detected, but there is circumstantial evidence and the case is intriguing.
- Could this simply be false positive HDV serology? Please discuss
- Page 4 (line 144) “No significant demographic differences were found between the two subgroups, except for age and risk factors for HIV co-infection [in which, as expected, subjects were more frequently younger and intravenous drug users 156 (IVDUs)].”
o Please clarify which “subjects” were younger and more frequently IDU
- Page 4 (line 157) “The two subpopulations were also comparable for what concerns the main clinical parameters, even though HCV mono-infected persons showed a trend towards some more liver damage, probably due to the fact that there was a lower proportion of subjects in this subpopulation who had already achieved a sustained virological response (SVR) at the study entry.”
o This is unexpected as generally HCV/HIV coinfection is thought to cause more severe liver disease than HCV monoinfection.
o How was the severity of liver damage measured (i.e. what metric are you basing this statement on)?
o The difference in SVR between groups was not statistically significant so you can only say there was a trend
o Do you think the (significant) difference in age may have contributed? i.e. have the older patients been infected for longer (and therefore had more time for liver damage)?
o There are significant differences for most of the blood parameters (LFTs, INR, albumin) – you should at least summarise this in your statement.
- Page 5 (table 1) Concerning “SVR after previous HCV treatments”, does that mean patients have been reinfected with HCV after previous clearance? You need to clarify as it is ambiguous and suggests you have patients in the cohort who have had SVR, yet they should not have detectable HCV RNA (by definition) unless they have been reinfected.
- Page 10 (line 352) “Interestingly, the anti-HD Co/S ratios observed were much higher in anti-HBc positive (median: 4.40) than in anti-HBc negative samples (median: 1.48, P < 353 0.01)”
o Please clarify what “anti-HD Co/S ratios” are and what they signify.
- The authors refer to evidence (papers) that HDV RNA can replicate in the livers of people who do not have HBV infection, but that it cannot produce infectious virus. Is it possible that the patient merely acquired HDV infection from her partner, which replicated for a while and produced immune response but not infectious particles, independent of HCV? Please discuss.
- Discussion is too long and waffly. Please condense and focus a bit more
Minor points
- On page 3 (lines 91, 92) a strange symbol is present. Should that be “&” or “+/-“ ?
- Page 4 (line 163) – strange symbol again
Comments on the Quality of English LanguageMinor English edits recommended after review by native English speaker
Reviewer 2 Report
Comments and Suggestions for Authors The authors presented a retrospective study of patients with co-infection with hepatitis Delta and C. Their data indicate the possibility of the transmission of HDV through HBV-unrelated viruses in humans. The study undoubtedly contributes to the development of our understanding of the transmission of hepatitis delta virus.Author Response
Please see the attachment.
